# Does COVID-19 Revamp Nurses' Compassion? Post-Pandemic Approach in Qatar

George Vellaramcheril Joy *, Kamaruddeen Mannethodi, Albara Mohammad Ali Alomari , Kalpana Singh , Nesiya Hassan , Jibin Kunjavara and Badriya Al Lenjawi

Nursing and Midwifery Department, Hamad Medical Corporation, Doha P.O. Box 3050, Qatar; kmannethodi@hamad.qa (K.M.); aalomari5@hamad.qa (A.M.A.A.); ksingh1@hamad.qa (K.S.); nesiyahassan@gmail.com (N.H.); jkunjavara@hamad.qa (J.K.); blenjawi@hamad.qa (B.A.L.)
* Correspondence: gjoy1@hamad.qa; Tel.: +974-5023-0815

**Abstract:** Aim: This study aimed to identify self-compassion among staff nurses after the COVID-19 pandemic in Qatar. Design: Descriptive cross-sectional survey design. Methods: Anonymous data were collected through an online survey using Microsoft Forms from 300 nurses in 14 health facilities in Qatar. The organization had almost 10,000 nursing staff working in different facilities. Data were gathered using a structured online questionnaire and included socio-demographic information, and the Self-Compassion Scale—Short Form was used to collect the remaining data. Correlation, *t*-test, and ANOVA analyses were conducted. Results: Nurses in the study showed high self-compassion. Among the sub-domain 'mindfulness', they showed comparatively high scores (7.96 ± 1.55), and the lowest score was for 'isolation' (6.15 ± 1.99). The score for 'self-kindness' was 7.29 ± 1.55, that for 'self-judgement' was 6.79 ± 2.01, that for 'common humility' was 6.62 ± 1.47, and that for the sub-domain 'over-identified' was 6.47 ± 1.91. Mindfulness scores were high among the nurse leaders. Moreover, over-identified scores were high among the nurses who were currently working under COVID-19 at the time of data collection. Conclusions: Nurses faced many difficulties while working during the COVID-19 pandemic, including a heavy workload and tension. The current study's findings add to our understanding of how COVID-19 affected the development of self-compassion. A rise in mindfulness, which aids nurses in managing stress at work and building resilience, further underscores an increase in nurses' acceptance of the COVID-19 pandemic. The findings also highlight how crucial it is to encourage self-compassion in individuals and offer them emotional support at such times, especially when there is a significant risk factor for mental health, such as COVID-19.

**Keywords:** post-pandemic; mental health; resilience; self-compassion; mindfulness; kindness; judgement; nurses



## 1. Introduction

The COVID-19 disease has become a dominant force, having a substantial impact on worldwide health and healthcare systems. During the COVID-19 pandemic, healthcare employees represented a vulnerable group who were subjected to direct contact with affected patients, an excessive workload, and experiences of physical exhaustion, fear, emotional disturbance, and sleep pattern dysregulation. Nurses were the first-line healthcare workers who cared for COVID-19 patients [1]. As front-liners, nurses witness the suffering and deaths of their patients, which affects their psychological well-being [2]. When confronted with difficulties or adversity such as a pandemic, self-compassion is highly significant, among other psychological factors [3]. Compassion can be described as an awareness of the suffering of others coupled with the motivation to alleviate and prevent this suffering [4]. Compassion involves sensitivity, recognition, understanding, emotional resonance, empathic concern, and tolerance for the distress generated by the suffering of others [5]. Self-kindness, common humanity, and experiencing the present without

over-identification with one's emotions are necessary qualities of self-compassion [4]. In this case, self-compassion enables us to appreciate other people's views and experiences by helping us to see our shared humanity, acknowledge that we all face challenges, and develop a balanced understanding of others' experiences [3].

Indeed, studies highlights that nurses' self-compassion is an important component that reflects their self-awareness and empathic understanding of their patients [6,7]. Through self-compassion, nurses may be particularly aware of the mindfulness and non-judgmental behavior approaches that can be used in their practice, as such approaches acknowledge that life can involve suffering at times, and that things may be beyond our control. Mindfulness approaches also help individuals to commit to living life in line with their own values and with awareness by avoiding over-identification with automatic thoughts and negative thoughts [8]. Evidence shows that self-compassionate people are less likely to develop negative self-evaluations and always have a strong sense of worth [9]. Additionally, self-compassion has a significant impact on the psychosocial abilities and behaviors of the general public, including happiness, good self-evaluation, and increased social connectedness [10]. Being mentally compassionate and understanding toward oneself when facing challenges or experiencing difficult moments is the act of practicing self-compassion, i.e., the capacity of a person to respect, care about, and be kind to others as well as to himself or herself [11]. Self-compassion is one of the newest topics of empirical interest among researchers, and it has emerged as a framework for employee mental health and well-being in both organizations and hospitals [12].

During challenging situations such as a pandemic, self-compassion is considered important to increase the protective factors of individuals in order to avoid mental health problems that may occur [13]. When confronted with difficulties or adversity such as a pandemic, self-compassion is highly significant, among other psychological factors [3]. Professionals engaged in helping others, including nurses, experience a lack of self-compassion due to working with patients who have experienced a traumatic event such as a pandemic [14]. More importantly, evidence shows that the post-pandemic era's high workload, job demands, and turnover are predictive of burnout and diminished well-being among healthcare professionals, including nurses. As a result, it is necessary to gauge the level of self-compassion among nurses in such periods [15].

In Qatar, few studies have been conducted on nurses after the COVID-19 pandemic (includes main two waves) to evaluate self-compassion aspects. This study aimed to explore the self-compassion of staff nurses after the COVID-19 pandemic. The research could aid in examining how the pandemic has affected nurses' ability to cultivate self-compassion.

*Theoretical Framework*

Shame resilience theory is the theoretical foundation for this study [16]. According to shame resilience theory, certain qualities can help people to understand and cope with difficult circumstances and quiet their inner critic, which is essential for wellbeing and promotes inner resilience [17]. At the beginning of the COVID-19 pandemic, nurses experienced dread, self-blame, and social isolation, but, as their experience progressed, they started to build support and meaningful relationships with people (such as families and coworkers), which made them resilient and empathic. They were able to develop self-compassion by moving along the shame resilience continuum from the fear zone to the growth zone [18].

## 2. Methods

### 2.1. Design

The study used a descriptive, cross-sectional research survey design. The study was conducted at the largest health organization in Qatar, which included 14 health facilities. The organization had almost 10,000 nursing staff working in different facilities.

## 2.2. Participants

The target population of the study was registered nurses working in the health organization in Qatar, including 14 health facilities. Based on the mean resilience score obtained from prior research (66.91 $\pm$ 13.34) [19], the sample size was calculated to be 268, with a target population of 10,000 nurses and a 95% confidence interval. Assuming a 12% non-response rate, a total sample size of 300 was determined. The sample calculation was performed as a part of the part of the work in [20]. The sample size (n) was calculated using the formula n = $Z^2 \times S^2 / (d)^2$, where Z = 1.96, S is the standard deviation, and d is the precision (d = Z $\times$ SOE). SOE is the standard error of the mean.

Participants in the research had to be licensed staff nurses with a minimum of one year of experience to meet the inclusion requirements.

## 2.3. The Instrument

Data were gathered using a structured online questionnaire. The demographic information of the subjects was included in the first section. Participants were asked a variety of demographic questions, including personal demographics such as gender, marital status, educational qualifications, years of experience, designation, and COVID-19 deployment status.

The Self-Compassion Scale—Short Form (SCS-SF), a five-point Likert scale with 12 questions ranging from 1 to 5, comprised the second section of the questionnaire. A score of 1 indicates 'almost never', while 5 indicates 'almost always'. The total score can be between 12 and 60. Higher results reflect greater levels of self-compassion. The measure also includes sub-domains for self-kindness, self-judgment, common humanity, isolation, mindfulness, and over-identified. Each item has two questions and a score ranging from 1 to 10 [21]. The scale showed good reliability in terms of Cronbach's $\alpha$ coefficient (0.84) and test–retest reliability (0.89 in the 2-week interval) [22].

## 2.4. Data Collection

The anonymized data were gathered via online questionnaires using Microsoft Forms during the third wave of COVID-19. The information sheet and survey link were sent to participants via the hospital staff nurses' emails. To boost response rates, a reminder email was sent every two weeks. Implied consent was used, where staff could refuse to participate in the survey by not returning their forms [23]. The researchers had no formal relationships with the participants.

## 2.5. Ethical Considerations

Institutional Review Board (IRB) approval was obtained from the Medical Research Centre (MRC-01-21-723). Emails with the questionnaires were sent to every employee at 14 healthcare facilities. Participants were made aware that their involvement was voluntary and that no personally identifiable information was collected. The study was conducted in full conformance with the principles of the Declaration of Helsinki and Good Clinical Practice (GCP) and within the laws and regulations of the Ministry of Public health, Qatar.

## 2.6. Data Analysis

A total of 300 subjects were enrolled. The distribution of participant data and the sample characteristics were determined using descriptive statistics. All statistical analyses were performed using the statistical packages STATA 17.0 and Epi-info (Armonk and Epi-info (Centers for Disease Control and Prevention, Atlanta, GA, USA)). STATA is a powerful statistical analysis and data visualization tool used in different fields by researchers. A score was calculated for self-compassion to include the responses of the nurses. STATA was used to summarize categorical data; frequencies and proportions were used. The unpaired t ANOVA test was used to evaluate quantitative data between two or more independent groups using STATA. All $p$ values < 0.05 were considered statistically significant and the presented $p$ values were two-tailed.

## 3. Results

### 3.1. Sample Characteristics

A total of 300 nurses participated in the survey. The mean age of the participants was 38.2 ± 7.2 years and more than one third (76%) of the participants were females. Most of the participants (42.7%) were in the age group of 35–44 years. The majority (76%) of the respondents were married and 74% were working as staff nurses. With regard to their experience, 26.3% of them had 1–3 years, 25.7% had 6 to 10 years, and 35% had more than 11 years. Most participants (72.3%) were graduate registered nurses, followed by charge nurses (14%), nurse educators (5%), chief nurses (6%), and executive and nursing directors (2%).

More than half of the nurses (60.3%) were assigned to COVID-19 facilities during the pandemic, and 21.7% were still working in COVID-19 facilities during the survey. The sociodemographic data of the sample are described in Table 1.

**Table 1.** Sociodemographic characteristics of the participants (*n* = 300).

| Factor | Level | Value (*n* = 300) |
|---|---|---|
| Age in years, mean (SD) | | 38.2 (7.2) |
| Gender | Male | 72 (24.0%) |
| | Female | 228 (76.0%) |
| Marital status | Single | 65 (21.7%) |
| | Married | 228 (76.0%) |
| | Divorce | 3 (1.0%) |
| | Widowed | 4 (1.3%) |
| Educational qualification | Diploma in Nursing | 29 (9.7%) |
| | BSN | 222 (74.0%) |
| | Master's degree and above | 49 (16.3%) |
| Years of experience in HMC | 1–3 years | 79 (26.3%) |
| | 4–5 years | 39 (13.0%) |
| | 6–10 years | 77 (25.7%) |
| | 11 and above | 105 (35.0%) |
| Designation | Charge nurse | 42 (14.0%) |
| | Executive/director of nursing | 8 (2.7%) |
| | Graduate registered nurse | 217 (72.3%) |
| | Head nurse | 18 (6.0%) |
| | Nurse educator/researcher | 15 (5.0%) |
| COVID-19 deployment status | Assigned before | 181 (60.3%) |
| | Currently working | 65 (21.7%) |
| | Never assigned | 54 (18%) |

### 3.2. Self-Compassion of Study Participants

The mean self-compassion score was 41.3 ± 5.91. The average score for the domain 'mindfulness' showed a comparatively high value (7.96 ± 1.55), and the lowest score was for 'isolation' (6.15 ± 1.99). The score for 'self-kindness' was 7.29 ± 1.55, that for 'self-judgement' was 6.79 ± 2.01, that for 'common humility' was 6.62 ± 1.47, and that for the sub-domain 'over-identified' was 6.47 ± 1.91. Tables 1 and 2 show the descriptive statistics of the study variables.

**Table 2.** Descriptive statistics of study variables (*n* = 300).

| Status | Mean | SD | Median; (IQR) Range |
|---|---|---|---|
| Self-compassion | 41.3 | 5.91 | 30.0 (26.0, 35.0) |
| Kindness | 7.3 | 1.55 | 7.0 (6.0, 8.0) |
| Judgement | 6.8 | 2.01 | 7.0 (6.0, 8.0) |
| Humanity | 6.6 | 1.47 | 7.0 (6.0, 8.0) |
| Isolation | 6.2 | 1.99 | 6.0 (5.0, 8.0) |
| Mindfulness | 7.9 | 1.55 | 8.0 (7.0, 9.0) |
| Over-identified | 6.5 | 1.91 | 6.0 (5.0, 8.0) |

*3.3. Association of Sociodemographic Variables with Study Variables*

The relationship between sociodemographic characteristics (age, gender, marital status, education, experience, designation, and COVID-19 deployment status) and self-compassion and its domains showed no statistical significance, except in terms of designation and COVID-19 deployment status. The mean score for the sub-domain 'over-identified' showed statistical significance (*p* = 0.02) among the nurses who were currently deployed (3.22) or assigned to a COVID-19 facility (3.33), compared with nurses never assigned to a COVID-19 facility (2.93). Moreover, 'mindfulness' among head nurses and executive/directors of nursing was high compared to other groups and was statistically significant (*p* = 0.002). Table 3 shows the associations between demographics and self-esteem, resilience, and self-compassion.

**Table 3.** Associations between demographics and self-esteem, resilience, and self-compassion.

| | | Self-Kindness, Mean (SD) | Self-Judgement, Mean (SD) | Common Humanity, Mean (SD) | Isolation, Mean (SD) | Mindfulness, Mean (SD) | Over-Identified, Mean (SD) | Self-Compassion Total, Mean (SD) |
|---|---|---|---|---|---|---|---|---|
| Gender | Male | 7.49 (1.44) | 6.97 (2.06) | 6.73 (1.68) | 5.96 (1.92) | 8.01 (1.57) | 6.28 (1.89) | 41.44 (5.77) |
| | Female | 7.23 (1.58) | 6.73 (2.00) | 6.59 (1.40) | 6.21 (2.01) | 7.95 (1.54) | 6.52 (1.92) | 41.24 (5.96) |
| | *p* value | 0.22 | 0.38 | 0.47 | 0.34 | 0.75 | 0.34 | 0.80 |
| Marital status | Single | 7.55 (1.60) | 6.89 (1.88) | 6.63 (1.36) | 6.11 (1.89) | 7.86 (1.63) | 6.40 (1.76) | 41.45 (6.23) |
| | Married | 7.19 (1.53) | 6.77 (2.04) | 6.63 (1.51) | 6.19 (2.01) | 8.00 (1.49) | 6.50 (1.95) | 41.28 (5.67) |
| | Divorce/separated | 8.66 (2.31) | 6.67 (4.16) | 6.67 (1.15) | 5.33 (3.51) | 7.0 (4.36) | 5.67 (3.21) | 40.00 (17.43) |
| | Widowed | 7.75 (0.5) | 6.25 (1.26) | 6.25 (0.96) | 5.50 (1.73) | 8.25 (0.5) | 6.25 (1.89) | 40.25 (3.77) |
| | *p* value | 0.13 | 0.92 | 0.97 | 0.79 | 0.64 | 0.87 | 0.96 |
| Education | BSN | 7.30 (1.51) | 6.66 (1.96) | 6.64 (1.46) | 6.12 (1.97) | 7.91 (1.45) | 6.34 (1.84) | 40.98 (5.55) |
| | Diploma | 6.76 (1.79) | 7.41 (1.94) | 6.65 (1.88) | 6.24 (2.32) | 8.03 (1.78) | 7.03 (1.88) | 42.14 (6.04) |
| | Master's degree/above | 7.53 (1.57) | 7.02 (2.24) | 6.53 (1.28) | 6.26 (1.92) | 8.16 (1.78) | 6.69 (2.20) | 42.20 (7.24) |
| | *p* value | 0.10 | 0.11 | 0.88 | 0.89 | 0.56 | 0.12 | 0.30 |
| Years of experience | 1–3 years | 7.34 (1.58) | 6.72 (2.01) | 6.59 (1.38) | 6.09 (2.16) | 7.99 (1.56) | 6.47 (1.89) | 41.20 (6.00) |
| | 4–5 years | 7.46 (1.65) | 6.59 (2.07) | 7.10 (1.19) | 6.03 (2.02) | 8.18 (1.45) | 6.05 (1.85) | 41.41 (6.23) |
| | 6–10 years | 7.23 (1.53) | 6.86 (2.01) | 6.54 (1.55) | 6.21 (1.92) | 7.91 (1.57) | 6.55 (1.94) | 41.30 (5.83) |
| | *p* value | 0.66 | 0.70 | 0.09 | 0.83 | 0.60 | 0.33 | 0.98 |
| Assigned to COVID-19 facility | Currently working | 7.31 (1.56) | 6.91 (2.01) | 6.77 (1.54) | 6.05 (2.07) | 8.08 (1.41) | 6.65 (1.98) | 41.55 (6.03) |
| | Assigned before | 7.25 (1.55) | 6.80 (2.02) | 6.60 (1.44) | 6.25 (1.90) | 7.90 (1.55) | 6.66 (1.91) | 41.46 (6.09) |
| | Never assigned | 7.41 (1.58) | 6.61 (2.01) | 6.53 (1.53) | 5.96 (2.19) | 8.04 (1.69) | 5.85 (1.74) | 40.41 (5.09) |
| | *p* value | 0.80 | 0.72 | 0.65 | 0.58 | 0.68 | 0.02 | 0.48 |

## 4. Discussion

Working in unusual circumstances may negatively affect self-compassion. The COVID-19 pandemic influenced the self-compassion of nurses. This study sought to explore the self-compassion of nurses during the COVID-19 pandemic. Nurses in the current study showed high self-compassion scores, which indicates that the nurses were less likely to fear judgment from others, increasing their willingness to help others and their ability to cope with difficult circumstances [24]. Self-compassion serves as a valuable coping resource when nurses experience negative life events. Nurses who are self-compassionate are less likely to catastrophize negative situations [25].

Self-compassion in the present study was higher than in other studies. A study conducted in Spain during the COVID-19 pandemic among nurses found that the mean score of self-compassion was 19.8 ± 4.4 [10]. The authors highlighted that self-compassion was developed as a psychologically protective factor that prevented nurses from experiencing stress and burnout. The pandemic could thus enhance mental health, lessen the negative effects of stressful events that may affect nurses, and promote self-compassion [26]. According to recent research on the self-compassion of nurses during the pandemic, self-compassion may enhance life satisfaction through effective coping [27]. Nurses in the study may have adopted positive coping mechanisms and adjusted them to the circumstances.

The nurses in the study showed a high level of mindfulness compared with other sub-domains of self-compassion. The result is consistent with another study conducted on nurses during the pandemic, experiencing high levels of mindfulness [28]. The authors stress that mindful nurses can observe their thoughts, emotions, and events without embellishing, denying, or repressing them. Mindfulness enables nurses to view the situation more broadly and involves a balanced response to uncomfortable feelings [7]. Evidence suggests that mindful nurses accept the present situation without being distracted by the future or past. The improvement in mindfulness scores highlights that nurses started to accept the COVID-19 pandemic, helping them to cope with stress at work and develop a sense of resilience [25].

However, isolation scores were low when compared with the other sub-domains. Studies conducted early during the pandemic highlighted that psychological strain involving perceived isolation from others at work contributed to poorer mental health among employees and played an important role in negative self-compassion attributes [7]. However, our study was conducted after the third wave of the COVID-19 pandemic, and the low isolation scores may indicate that the nurses overcame the feeling of isolation and became mindful. The low isolation scores are incongruent with another study conducted on nurses' mindfulness and self-compassion training among nurses [29]. By forming strong social bonds with other people, the author argues that nurses' feelings of loneliness are lessened, enhancing both their ability to perform their duties and their interactions with patients [29].

While considering the associations, mindfulness among head nurses and executive/directors of nursing was high compared to the staff nurses who participated in the study. Mindfulness is a wellness strategy that is inversely proportional to stress and burnout; high stress and pressure will decrease the mindfulness levels in individuals [30]. Previous evidence highlighted that nurses who work on the front line face more challenges and stress [2] compared with leaders, which in turn can reduce mindfulness. Increased mindfulness was associated with decreased feelings of isolation [31,32]. The high mindfulness scores and low isolation scores among nurses during the third wave of the pandemic in the current study could be explained by the participants' previous experience during the first and second waves, in which they developed an awareness of COVID-19 and developed empathy with their patients and colleagues.

Moreover, over-identified scores were high among the nurses who were currently working or assigned to a COVID-19 facility, compared with nurses who were never assigned to a COVID-19 facility. Over-identification leaves nurses immersed in their current emotional reactions and leads to rumination on their difficulties, which leads to burnout. Evidence shows that burnout among nurses was a crucial issue during the COVID-19

pandemic [33]. Studies conducted on nurses' burnout and self-compassion identified that burnout has a significant and positive correlation with over-identification [34]. The study result points out that even during the third wave, nurses had not completely recovered from their burnout and stress.

### 4.1. Implications for Practice

Nursing care must include compassion as a vital component. Examining the compassion that nurses may feel for themselves is a crucial task, because, without the capacity for self-compassion, nurses may not be well-equipped to demonstrate compassion toward the patients that they care for. The findings of this study may suggest that many nurses have high levels of self-compassion. However, on every subscale item, there were nurses who responded negatively and had very low scores (i.e., they showed less common humanity and greater over-identification). In terms of nurses' self-compassion, further study is required. It is crucial for practice to recognize and assist those nurses who lack self-compassion.

### 4.2. Limitations

Convenience sampling was used to perform the study, from 14 different hospitals, which were all located in Qatar, which limits the generalization to other institutions. Due to the study's use of an online questionnaire, there may have been some bias in the reporting, i.e., the propensity of participants to present a more positive view of themselves, meaning that self-reporting surveys may introduce bias due to social desirability [30]. Participants may fabricate answers to conform to socially acceptable standards or avoid criticism, or they may falsify the information that they report (self-deception) [19].

### 4.3. Recommendations

The study's findings imply that nurses may exhibit a high degree of self-compassion. There is also a need for additional research on how to increase self-compassion. Studies that compare the relationship between self-compassion and other psychological variables may provide insight into the processes involved developing one's self-compassion. More qualitative research is required to understand the circumstances that give rise to compassion.

### 5. Conclusions

Nurses faced many difficulties while working during COVID-19, including a heavy workload and tension. However, pandemics might make nurses more compassionate toward themselves. The results of the current study add to the body of knowledge on how COVID-19 affected the development of self-compassion. As the pandemic progressed, nurses gained psychological strength from their self-esteem and self-compassion via their psychological resilience. Increased scores for mindfulness indicated their acceptance of their current circumstances without dwelling on the past or the future. A rise in COVID-19 pandemic acceptability was highlighted by an improvement in mindfulness, which aided the nurses in coping with work-related stress and building resilience.

The findings also highlight the need to increase people's levels of self-compassion and provide them with emotional support during such times, especially when there is a substantial risk factor for mental health, such as COVID-19. Healthcare organizations should initiate psychological interventions to boost individuals' self-compassion for themselves, which can alleviate mental health problems during crisis periods.

**Author Contributions:** G.V.J., A.M.A.A., K.S., N.H., K.M., J.K. and B.A.L. each participated sufficiently in the work to take public responsibility for appropriate portions of the content. All authors have read and agreed to the published version of the manuscript.

**Funding:** Open access funding provided by the Qatar National Library.

**Institutional Review Board Statement:** Institutional Review Board (IRB) approval (MRC-01-21-723) was obtained from the Medical Research Centre (MRC). The study conducted in full conformance

with principles of the "Declaration of Helsinki", Good Clinical Practice (GCP) and within the laws and regulations of Ministry of public health in Qatar.

**Informed Consent Statement:** Not applicable.

**Data Availability Statement:** Data available on request due to privacy/ethical restrictions.

**Acknowledgments:** The authors would like to thank all nurses who participated in this study for their unique contributions. The authors acknowledge the medical research center for their review and approval to conduct the study.

**Conflicts of Interest:** The authors declare no conflict of interest.

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
