# Peer review of "Does COVID-19 Revamp Nurses’ Compassion? Post-Pandemic Approach in Qatar"

_covid, doi:10.3390/covid4080087_

Round 1
Reviewer 1 Report
Comments and Suggestions for Authors
The authors present a good research paper.
- The relevance of the topic: Good.
- Abstract: Can be improved.
- Keywords: Can be improved.
- Introduction: Can be improved.
- Methodology: Can be improved.
- Results: Good.
- Discussion: Good.
- Conclusions: Can be improved.
However, ACCEPT AFTER MINOR REVISION. In general, the paper follows an adequate structure and correct scientific support and can be published considering some limitations. The study is interesting in the field of Covid 19. However, there are a series of limitations that should be considered.
In the first place, carry out a review of the existing literature related to the subject, being essential to inquire into the MPDI – COVID journal itself, since there are papers related to its manuscript that can help to improve it. Therefore, include those references, if any, especially from the last five years. In addition, recommend reading some papers related to the topic of Covid 19.
Cruz, R., Diz-de Almeida, S., López de Heredia, M., Quintela, I., Ceballos, F. C., Pita, G., ... & Zeberg, H. (2022). Novel genes and sex differences in COVID-19 severity. Human molecular genetics, 31(22), 3789-3806.
Mantovani, A., Morrone, M. C., Patrono, C., Santoro, M. G., Schiaffino, S., Remuzzi, G., ... & Covid-19 Commission of the Accademia Nazionale dei Lincei Cappuccinelli Pietro 13 Fitzgerald Garrett 14 Bacci Massimo Livi 15 Melino Gennaro 16 Parisi Giorgio 17 Rappuoli Rino 18 19 Rezza Giovanni 20 Vineis Paolo 21. (2022). Long Covid: where we stand and challenges ahead. Cell Death & Differentiation, 29(10), 1891-1900.
Specific comments.
Title. The title of the manuscript is correct.
Abstract. Incorporate in the summary, a more precise sentence of the methodology and the results.
Keywords. Use keywords other than those used to formulate the manuscript title.
Introduction. This section presents the problem in a coherent and clear manner with the correct support of the scientific literature. However, it is convenient to update the references, since there are different documents related to the subject and no mention is made, and it would even be interesting to mention the different existing studies related to Covid 19. Also, it could be a future study of review. Some bibliographical references are attached to carry out the section of Covid 19 and post pandemic:
Agag, G., Aboul-Dahab, S., Shehawy, Y. M., Alamoudi, H. O., Alharthi, M. D., & Abdelmoety, Z. H. (2022). Impacts of COVID-19 on the post-pandemic behaviour: The role of mortality threats and religiosity. Journal of Retailing and Consumer Services, 67, 102964.
Zancajo, A., Verger, A., & Bolea, P. (2022). Digitalization and beyond: the effects of Covid-19 on post-pandemic educational policy and delivery in Europe. Policy and Society, 41(1), 111-128.
Zhou, T., Xu, C., Wang, C., Sha, S., Wang, Z., Zhou, Y., ... & Wang, Q. (2022). Burnout and well-being of healthcare workers in the post-pandemic period of COVID-19: a perspective from the job demands-resources model. BMC Health Services Research, 22(1), 1-15.
Methods. Modify the method section, and specifically, in the section: Design.
- Study design. To write the design section, we recommend that you take some of the following methodologists as references.
Ato, M., López-García, J. J., & Benavente, A. (2013). A classification system for research designs in psychology. Annals of Psychology, 29(3), 1038-1059.
Montero, I., & León, O.G. (2007). A guide for naming research studies in psychology. International Journal of Clinical and Health Psychology, 7(3), 847-862.
Results. Summary of study data and table are correct.
Discussion. The section Discussion is correct.
Conclusion. Differentiate the discussion of the main conclusions of the study. To do this, you must create this section. And modify the limitations of the study and locate them in said section at the end. Also, they must be direct, and highlight the main contributions of the study.
References. They should be reviewed and updated according to the publication standards. There are many errors in the references. Therefore, correct them and adapt them to the magazine's regulations.
Comments on the Quality of English LanguageThe authors present a good research paper.
- The relevance of the topic: Good.
- Abstract: Can be improved.
- Keywords: Can be improved.
- Introduction: Can be improved.
- Methodology: Can be improved.
- Results: Good.
- Discussion: Good.
- Conclusions: Can be improved.
However, ACCEPT AFTER MINOR REVISION. In general, the paper follows an adequate structure and correct scientific support and can be published considering some limitations. The study is interesting in the field of Covid 19. However, there are a series of limitations that should be considered.
In the first place, carry out a review of the existing literature related to the subject, being essential to inquire into the MPDI – COVID journal itself, since there are papers related to its manuscript that can help to improve it. Therefore, include those references, if any, especially from the last five years. In addition, recommend reading some papers related to the topic of Covid 19.
Cruz, R., Diz-de Almeida, S., López de Heredia, M., Quintela, I., Ceballos, F. C., Pita, G., ... & Zeberg, H. (2022). Novel genes and sex differences in COVID-19 severity. Human molecular genetics, 31(22), 3789-3806.
Mantovani, A., Morrone, M. C., Patrono, C., Santoro, M. G., Schiaffino, S., Remuzzi, G., ... & Covid-19 Commission of the Accademia Nazionale dei Lincei Cappuccinelli Pietro 13 Fitzgerald Garrett 14 Bacci Massimo Livi 15 Melino Gennaro 16 Parisi Giorgio 17 Rappuoli Rino 18 19 Rezza Giovanni 20 Vineis Paolo 21. (2022). Long Covid: where we stand and challenges ahead. Cell Death & Differentiation, 29(10), 1891-1900.
Specific comments.
Title. The title of the manuscript is correct.
Abstract. Incorporate in the summary, a more precise sentence of the methodology and the results.
Keywords. Use keywords other than those used to formulate the manuscript title.
Introduction. This section presents the problem in a coherent and clear manner with the correct support of the scientific literature. However, it is convenient to update the references, since there are different documents related to the subject and no mention is made, and it would even be interesting to mention the different existing studies related to Covid 19. Also, it could be a future study of review. Some bibliographical references are attached to carry out the section of Covid 19 and post pandemic:
Agag, G., Aboul-Dahab, S., Shehawy, Y. M., Alamoudi, H. O., Alharthi, M. D., & Abdelmoety, Z. H. (2022). Impacts of COVID-19 on the post-pandemic behaviour: The role of mortality threats and religiosity. Journal of Retailing and Consumer Services, 67, 102964.
Zancajo, A., Verger, A., & Bolea, P. (2022). Digitalization and beyond: the effects of Covid-19 on post-pandemic educational policy and delivery in Europe. Policy and Society, 41(1), 111-128.
Zhou, T., Xu, C., Wang, C., Sha, S., Wang, Z., Zhou, Y., ... & Wang, Q. (2022). Burnout and well-being of healthcare workers in the post-pandemic period of COVID-19: a perspective from the job demands-resources model. BMC Health Services Research, 22(1), 1-15.
Methods. Modify the method section, and specifically, in the section: Design.
- Study design. To write the design section, we recommend that you take some of the following methodologists as references.
Ato, M., López-García, J. J., & Benavente, A. (2013). A classification system for research designs in psychology. Annals of Psychology, 29(3), 1038-1059.
Montero, I., & León, O.G. (2007). A guide for naming research studies in psychology. International Journal of Clinical and Health Psychology, 7(3), 847-862.
Results. Summary of study data and table are correct.
Discussion. The section Discussion is correct.
Conclusion. Differentiate the discussion of the main conclusions of the study. To do this, you must create this section. And modify the limitations of the study and locate them in said section at the end. Also, they must be direct, and highlight the main contributions of the study.
References. They should be reviewed and updated according to the publication standards. There are many errors in the references. Therefore, correct them and adapt them to the magazine's regulations.
Author Response
thank you for the comments , kindly find the attached

Reviewer 2 Report
Comments and Suggestions for Authors
1. Please add more relevant references from MDPI, for example:
- Self-Compassion, Work Engagement and Job Performance among Intensive Care Nurses during COVID-19 Pandemic: The Mediation Role of Mental Health and the Moderating Role of Gender
- COVID-19 Infection among Nursing Students in Spain: The Risk Perception, Perceived Risk Factors, Coping Style, Preventive Knowledge of the Disease and Sense of Coherence as Psychological Predictor Variables: A Cross Sectional Survey
- Determinants of COVID-19 Vaccine Uptake and Hesitancy among Healthcare Workers in Tanzania: A Mixed-Methods Study
2. Methodology part can be more extended. More information about data collection and sample size and how you calculated the sample size. More information how stata helped you in analysis. You can add some plots about results as well.
3. Conclusion part can not support the paper please extend conclusion part and provide a better explanations.
Author Response
thank you for the comments

Round 2
Reviewer 2 Report
Comments and Suggestions for Authors
The paper is accepted in current version.